# Workplace Structures and Culture That Support the Wellbeing of People with an Intellectual Disability

**DOI:** 10.3390/ijerph21111453

**Published:** 2024-10-31

**Authors:** Andrew Joyce, Perri Campbell, Jenny Crosbie, Erin Wilson

**Affiliations:** Centre for Social Impact, School of Business, Law and Entrepreneurship, Swinburne University of Technology, Hawthorn 3122, Australia; pcampbell@swin.edu.au (P.C.); jcrosbie@swin.edu.au (J.C.); ewilson@swin.edu.au (E.W.)

**Keywords:** workplace culture, organisational structure, wellbeing, intellectual disability

## Abstract

There is little research on health-promoting workplace settings focused on people with an intellectual disability. There are a range of supported and open employment workplaces where people with an intellectual disability work, and this is an important setting that can influence health and wellbeing outcomes. The health promotion research that has been conducted with people with an intellectual disability has been programmatic in focus and lacks a broader settings and ecological perspective. This paper reports on analysis conducted across four organisations that employ people with an intellectual disability and included 47 in-depth interviews conducted with staff and supported employees. The aim was to examine the organisational characteristics, structures, and cultural elements that contribute to positive wellbeing. The key elements were offering a diversity of roles and opportunities, customised training and task matching, a flexible approach to work rosters, offering a range of workplace environments (e.g., busy versus calm environments), and providing holistic and tailored support. The results illustrate that providing positive wellbeing outcomes in the workplace for this cohort cannot be considered from a program perspective but as a whole of organisation design and culture. With the current movement towards more opportunity in open employment, it will be important that these features are replicated in all workplaces where people with an intellectual disability are employed. Further research and policy work is required for this ambition to be realised.

## 1. Introduction

Determining the number of people with an intellectual disability is difficult due to changing measurement criteria and classifications [1]. The traditional approach was based solely on the standardised measurement of an IQ test; however, criteria now include limitations in daily activities [1]. In Australia, estimates vary from 1.6% of the population to 2.7% of the population depending on which criteria are used [1], and the estimated prevalence for the entire world is 1% [2]. In respect of health and wellbeing, this population cohort has a higher rate of avoidable mortality [3], and an increased rate of mental health problems [4]. While this population cohort has been shown to have poorer health relative to the general population, they are less likely to be the recipients of health promotion programs [5], and there has been less health promotion research with this group [6]. The research that has been conducted has been mainly limited to small programs with far less focus on policy and settings approaches [7,8,9]. A number of researchers have recommended more research on settings-based approaches for promoting health and wellbeing for people with an intellectual disability [7,9,10,11,12]. Given that health-promotion approaches that are directed at the whole population do not reach people with an intellectual disability [8,9], it is imperative that there is guidance on how to promote health and wellbeing through key settings for this population cohort.

One of the key settings through which to promote health and wellbeing for people with intellectual disability is workplaces [12]. There has been very little research on how to promote health and wellbeing in workplaces for this population cohort [12]. This paper will examine four case study organisations in how they intentionally design a workplace that promotes wellbeing. The study includes analysis of in-depth interviews that were conducted with 47 people (16 supported employees, 24 staff and managers, 6 external employers, and one family member). This was part of a larger study examining employment transitions between supported and open employment and their health and wellbeing experiences in supported employment. The questions and analysis of this paper focused on the organizational structures, processes, and culture that enable these organisations to address the health and wellbeing of supported employees. The particular focus of analysis was the perceptions of staff and managers on how organisational structures and processes influenced health and wellbeing, and how supported employees themselves viewed the employment experience. Health and wellbeing are terms that are often used interchangeably [13]. The WHO uses wellbeing in its definition of health which encompasses physical, social and mental wellbeing [13]. Wellbeing itself also includes the dimensions of quality of life [13]. Thus, we used the terms together to provide the broadest possible conceptual approach which enabled the interview participants themselves to provide their perceptions of how to support health and wellbeing (quality of life) within the workplace context. While the intent of the interviews was to cover all aspects of health, the interview participants mainly covered mental and social wellbeing which are the main focuses of this paper.

### Workplace and Wellbeing for People with an Intellectual Disability

There has been considerable debate on inclusive employment options for people with intellectual disability but less focus on wellbeing within the workplace. Inclusive employment is characterised by being integrated into the community and including both people with and without disability, which can typically be found in open employment and social enterprises [14]. This contrasts to sheltered employment where people only work with other people with intellectual disability typically for less than award wages on production-line-type work [14]. Over the last few decades in Australia, and mirrored internationally, there have been changes with respect to employment policy. The shift has been from segregated employment options, such as sheltered workshops, to the offering of, theoretically, more inclusive options. In Australia, from the 1990s, organisations employing large numbers of people with intellectual disability were labelled as disability ‘business services’ or ‘business enterprises’ and from 2008 were called Australian Disability Enterprises (ADEs). Figures from 2022 showed that there were 477 ADEs, and they employed 16,000 people [15]. The transition from ADEs to open employment is very low, with 4% of 15–24 year olds and 1% of people aged over 25 years transitioning from ADEs to open employment [16]. The Disability Royal Commission presented split views on whether ADEs should be discontinued or reformed [17]. Some ADEs are now self-referring as social enterprises; however, the Commission did not think that they were sufficiently inclusive, lacked a diverse workforce, and did not provide adequate training opportunities to promote transition to open employment. Further, it was expressed that people in these settings are still at risk for exploitation, violence, and abuse [17]. Low transition rates between supported and open employment are also found internationally [18]. People with an intellectual disability have voiced their preference for inclusive employment opportunities through social enterprise and open employment settings [19,20] and the Disability Royal Commission agreed that these should be the first options considered from a rights based perspective [17].

While it is recognised that workplaces for people with intellectual disability need to be inclusive, there has been very little focus on how to promote health and wellbeing for this group within workplaces [12]. There is a well-established evidence base for workplace health promotion. Health promotion interventions that include a range of strategies spanning both organisational change actions and education can be effective in improving health and wellbeing indicators [21,22,23], whereas education-only interventions are less effective [24,25]. Reviews of interventions have found that multi-component programs are effective in improving mental health, nutrition, and physical activity [21,22,23,24,25,26]. These are all important topics in improving the health of people with intellectual disability, and yet there is scant research addressing this setting for this group. More broadly, one of the criticisms of workplace health promotion is the gap in research with different population groups [27].

Another criticism of workplace health promotion is the focus on intervention approaches rather than consideration of how the design and operation of the workplace itself impacts on health and wellbeing [28]. One exception is some work on the role of how managers support workplace wellbeing for employees with an intellectual disability [29]. The focus of this research was on how health promotion is delivered in a workplace, so it was somewhat programmatic in perspective. However, the results did illustrate some themes related to the culture and structure of the organisational case study. These themes included constant processes of ‘checking in’ with employees on their experiences, flexibility in approaches to suit different workplaces, and a strong focus on empowerment and relationship building between managers and employees [29].

There is a need for more research on the organisational structure and culture that supports the health and wellbeing of people with intellectual disability in the workplace. The findings of this type of research can help in developing a guide that can support all workplaces that employ people with an intellectual disability [12]. Understanding how the organisational design and culture impacts on health and wellbeing rather than just health promotion program delivery is important in being able to design the best possible workplaces [30]. To address this research gap, a case study approach was used to understand from staff and managers what they considered were the key structures and processes that supported the health and wellbeing of people with an intellectual disability in the workplace. Further, interviews with people with intellectual disabilities themselves were undertaken to gain their perspective on what they valued in the workplace. The aims of this research were to examine what are the organisational structures, processes, and culture that can enable an organisation to positively impact on the health and wellbeing of people with an intellectual disability.

## 2. Method

The data analysed came from a project funded through the Information Linkages and Capacity Building Scheme of the Department of Social Services. The aims of the research project were to examine the specific features of supported employment organisations that were able to successfully transition people to open employment opportunities. Part of this research agenda focused on examining the organisational features and structures that supported inclusion and wellbeing. A case study approach was chosen based on the depth of analysis required to examine how the culture, structure, and operation of an organisation could impact on health and wellbeing [31,32].

Four organisations participated in the study. Each organisation provided a range of employment options within their organisation including more traditional supported employment typical of disability enterprises and more community facing roles but still with a range of job supports provided. The four organisations were specifically selected as they had a history of supported employees transitioning to open employment, which is rare among disability enterprises as transition rates are very low. This means these organisations were operating more like social enterprises with community facing roles which also makes them interesting for this research given the reputation of social enterprises in supporting health and wellbeing [14]. Forty-seven semi-structured interviews were undertaken with participants to understand how the organisations supported health and wellbeing (16 supported employees, 24 staff and managers; 6 people from partner organisations and 1 family member). One organisation had been involved in the project for three years, and the other three organisations only one year. Most of the data collection took place with organisation 1 (33 interviews), 7 interviews from organisation 2, 5 interviews from organisation 3, and 2 interviews from organisation 4. Some interviews were conducted face to face at the location of the workplace, but the majority were conducted online.

The inclusion criteria for an interview were staff and managers of people in the workplace, employees with an intellectual disability, family members, and open employers with experience of employing people with an intellectual disability. Inclusion criteria for supported employees was experience in attempting open employment and verbal abilities to take part in an interview. The research team were not privy to the official diagnosis of the interview participants, but the majority of the supported employees do have intellectual disability and neurodiverse characteristics. People were informed about the project during team meetings and were told to contact their manager if they wished to participate. Supported employees then discussed the interviews with their manager and if they agreed a time was organized with the researcher. Staff and managers directly contacted the researcher if they were interested in participating. Participants were provided with a plain language statement, which was also available in Easy English, and if required, the information was read to participants.

A semi-structured question schedule was developed that focused on two key elements, the processes in transitioning from a supported to open employment context, and secondly, the health and wellbeing supports provided in the workplace. This paper analysed data from the second of these topics. The interview questions for staff and partner organisations specifically covered the types of health and wellbeing supports provided and how this was related to organizational characteristics. Example questions were as follows:How do you support health and well-being in the workplace?Do you think there is a healthy work life balance? In what way?How do you support physical and mental health of people with a disability at work?How do you think the structure and culture of the organisation impacts on health and wellbeing?

The questions for people with an intellectual disability were more general and not as specific to organisational characteristics to suit the ability levels of this participant group. Example questions were as follows:What is your job?What do you like about your job?What skills do you need to do this job?What supports do you need to do your job?How do you feel at work?

Questions for the family member focused on the supports in the workplace they considered necessary for successful employment outcomes and good health and wellbeing. However, this interview mainly focused on the employment transition aspect of the study, and their data were not analysed for this study. There was analysis of the partner organisations’ data, as they related to the topic of this paper, but the data from the family member was only relevant to the employment transition part of the larger study and has not been included in this paper. Interviews were on average one hour in duration, and all interviews were recorded and professionally transcribed.

Given the limited research on health and wellbeing in the workplace for people with an intellectual disability, an inductive thematic analysis approach was undertaken [33]. According to best practice qualitative research, each of the data collection and analytic steps have been described [34,35,36]. To reduce bias and increase the confidence of the findings, a method of researcher triangulation was used where two authors were involved in both conducting the interviews and coding the data [37]. The data were first analysed using thematic analysis to uncover themes related to organisational factors that influenced health and wellbeing outcomes [38,39]. This involved a process of reading the data line by line and grouping the data into meaningful categories. The next analysis stage involved drawing connections between the key themes [35] and how these different organisational features combined together to impact the health and wellbeing of employees. These final themes were then discussed and refined with all study authors. As a credibility and integrity check, the themes of the study were presented and discussed in in a meeting with two staff representatives from each of the four organisations who had been part of the steering committee of the project [36]. The study was approved by the Human Research Ethics committee of Swinburne University of Technology.

## 3. Results

The following sections present the main organisational characteristics that support health and wellbeing of people with an intellectual disability in the workplace. These themes are as follows: diversity of roles and opportunities; flexible approach to work rosters; customised training and task matching; accessible and modifiable layout; offering a range of workplace environments (e.g., busy versus calm environments); providing holistic and tailored supports; and prioritising social connections and respect. The data are mainly drawn from the interviews that were conducted with staff, managers, and supported employees, with some data from partner organisations also analysed. The data from the family member was not included. Much of the data from the partner organisations and the family member were more relevant for the other aspect of the larger study on employment transitions between supported and open employment.

### 3.1. Diversity of Roles and Opportunities

The organisations provided a number of different work sites and roles which helped support positive wellbeing. According to staff, having diverse work sites and roles means that participants can change their work roles to support productivity and attain better mental health outcomes:
*“And informally, they’re [disability enterprise supervisors are] tuned into the precursors for anything that might happen; if the client’s having a down day, or their mood is low, or whatever. And they’re very tuned in and mindful of that. And then can…change work roles around, so there’s a lot of flexibility. And informally, they might talk to the client around, hey, maybe you need to go home and just rest today and then come back tomorrow, or whatever. So that’s the essence, or one of the crucial essences of supported employment”.*(Organisation 2, staff 18)

Diversity of opportunity is achieved by having multiple sites, different business streams, and different jobs roles, even on one site. People are enabled to access these/move around/transfer and gain diverse experiences. The organisations are structured in such a way as to have diverse work opportunities to maximise chances to test out preferences and interests, and learn new skills. Some of the case study organisations have multiple business streams on one site and also run a labour hire arm of the business that enables experience on different work sites.
*“I learnt there was so many different things to do, my aim was to learn to be able to do everything in the kitchen or in the canteen, so that I could be thrown anywhere. Like even today, I was serving customers, and then all of a sudden we needed to make some potato cakes…And then [name] got stuck on the coffee machine with customers, and I finished already putting the customer’s coffee through the till…and there’s been other times where it’s been the other way around, I’ve been stuck on the coffee machine and Chris has finished off my job…The variety is probably right up there. But also being front of house, that’s my main role. And the reason why I like front of house is because you’re dealing with the customers, and you feel like you’re actually helping customers, like you’re doing a service. And the customers are great. They’re polite, they’ll have a joke with you—yeah, they’re just really good, and very respectful”.*(Organisation 1, Supported Employee 2)
*“Every day is different. Mostly I do serve customers. Everything is different. Sometimes we have big jobs like folding and sometimes we don’t. Yeah, it just depends on the week and the day… it gives us a good feeling that we can do a lot more”.*(Organisation 2, Supported Employee 14)

According to staff and managers of the different organisations, being provided with a variety of different work roles provides people with confidence that they can handle change in the workplace, which increases their resilience and supports their mental health. This change could include having to perform different roles, or working with different people, being able to work in different environments, or being able to learn how to handle different equipment. However, managing change and its frequency in the workplace requires planning and support. In supporting change, Disability Enterprise staff ensure that there has been plenty of discussions with supported employees about their work preferences and what kinds of change they might like to attempt. Change is managed through allowing individuals to have time to get comfortable in new roles and environments, with support provided to assist this.

### 3.2. Flexible Approach to Work Rosters

Related to flexibility of roles, support staff also ensured that there was flexibility with respect to work hours. This was determined in consultation with the supported employee, family and service providers. It was a key principle that supported employees had autonomy in being able to choose their work allocation and roles. According to interview participants, it was also important that work hours were coordinated against other elements of someone’s life to ensure there was balance:
*“I chose how many days I would like to do… if you’ve got back problems or something that you’re finding it hard to do a full day, some people can do a half day”.*(Organisation 1, Supported Employee 9)
*“I think a perfect balance would look like maybe three days a week of work mixed in with some social activities because [Name] really thrives when he’s with others and he works really well when he’s around other people. He really enjoys that aspect and he can get low when he’s not around other people”.*(Organisation 2, staff 17)

A flexible approach to work hours and tasks meant the person felt secure and comfortable in the workplace and had a sense of control over their hours and roles. Further, there was a conscious approach to ensuring that there was a balance between work roles and other life pursuits. This necessitated good relationships between different service providers and those responsible for organising funding plans.

### 3.3. Customised Training and Task Matching

Providing customised training and task matching helps to promote confidence in the workplace and a sense of accomplishment. Customised training and development involves a range of approaches including certified training opportunities, hands-on learning for specific work tasks, and development of soft skills such as interacting with peers. Management and staff work together on tasks with supported employees to provide job coaching. This helps direct which work tasks and training opportunities need to be provided. According to interview participants, this tailored and collegial approach to job training was seen as another important element of the workplace for staff and employees:
*“He’s a great supervisor; he’s basically taught me everything about the job, what areas are what and how to do it properly, which I guess would vary from supervisor to supervisor probably. He’s taught me in a way that I understand, and I appreciate, and I agree with as well”.*(Organisation 1, Supported Employee 8)

The process of task matching involves staff working closely with supported employees to understand their interests and skills and to find tasks and industries that align with their strengths. For some supported employees, this may involve a task with a very specific focus. Some supported employees find social environments challenging and they prefer to work in environments where there is less social interaction such as a laundry or nursery (greenhouse). Each of the case study organisations were large, which carries the advantage of being able to provide a range of opportunities that can be matched to individual skills and interests. Having such a large array of options means that supported employees can work in areas that match their interests and abilities while also developing skills that are necessary to succeed in their preferred vocation. As an example, one supported employee had a particular interest in hairdressing but also required some specific training to develop the communication requirements of this role:
*“She could do the hairdressing and makeup, but she had a lot of barriers around her interactions, communication skills and managing that side of things due to her autism. So she practised in some of our groups to begin with and learnt some skills”.*(Organisation 1, Staff 10)

While providing customised tasks was seen as pivotal, it was also mentioned by interview participants that a strong sense of purpose of the role itself was important. That is, the task itself was not merely used as a training exercise or to occupy time; there needed to be some work-related purpose intrinsic to business operations.
*“Oh I think give them something that we do need to have, not just “you stand over there and rip the piece of paper to 50 pieces for me every day”. It needs to be something that they know we need to have happen and then they know that they’re contributing and they know that it is well appreciated, that what they’re doing is a good job”.*(Organisation 2, employer 1)
*“I take a lot of pride in what I do…To know that they can rely on me to do a good job, which means they don’t have to look over my shoulder”.*(Organisation 3, Supported Employee 15)

This process of customised approach to training and task matching is instrumental in promoting the confidence and wellbeing of employees.

### 3.4. Modifying Work Practices, Layout and Accessible Equipment

Modifying work practices, layouts and having accessible equipment is another key element of the operation of the organisations. Each organisation has several areas and rooms where the layout can be modified to suit different jobs and different teams of employees. This can involve moving furniture and machinery. Some of the ‘reasonable adjustments’ that supported employees require can be very simple, such as the example of laminated instruction cards attached to machinery and equipment. An example of a simple modification was using coloured stickers on a cash register to help a supported employee:
*“She’s put sheets out for me to know the basic stuff to read which is a café sandwich, or a basic sandwich or a wrap that comes to the till. With the coffees, there’s buttons to press, she’s put a sticker on the buttons for small and large … And it does make it more enjoyable, and more of a happier workplace. But it also makes it an efficient workplace as well”.*(Organisation 1, Supported Employee 2)

As this employee describes, the modification provided a sense of support and confidence to be able to perform her role successfully which increased her sense of wellbeing in the workplace.

### 3.5. Offering a Range of Workplace Environments

Closely related to modifying the layout and providing a variety of tasks is providing a range of different workplace environments. This refers to providing a range of spaces that can accommodate both social activity and times for solitude. This is different to just having a variety of tasks: it relates to having a variety of specific physical spaces in the workplace to accommodate different needs. Again, the advantage of having a diverse organisation is that some areas can be socially active and busy (i.e., the café) and others (i.e., the laundry) provide a calmer pace. According to interview participants, it was considered important that supported employees have the choice of experiencing a calm or busier workplace environment and that these needs can change over time or even each day:
*‘And you just go to the supervisor, “Can I just have some time out?” And they understand, “Okay, all right”. Then they’ll put you somewhere, just quietly on your own and you’ll just need some chill out time. Because sometimes you just want to work on your own’.*(Organisation 1, Supported Employee 4)

Quiet rooms designed for rest and breaks were made available to supported employees. The purpose of these available/free rooms was to enable areas where people can rest, have some privacy, and also a place for personal conversations. The rooms were an important empowerment strategy where people felt able to choose how they spent their time:
*“When we have break, any free room, any free function room, we just go in. Sometimes I go to the reception if there’s some people in those ones. I’ll just go, well, what rooms are free, and there’ll be a room, and it’s great because they’re dead quiet and you just sit in there”.*(Organisation 1, Supported Employee 3)
*“When I’m on lunch break, I just put my headphones on, watch my laptop and just veg out. You probably need to just have that time where you’re just doing that”.*(Organisation 1, Supported Employee 5)

Similar to each of the themes to date, the purpose is to create an accommodating and highly agile workplace environment that is inclusive and provides confidence to the supported employee in the workplace. This ability to manage their time in different spaces and with varying levels of social connection was viewed as particularly important for promoting positive mental health.

### 3.6. Holistic and Tailored Supports

The staff were focused on ensuring that a holistic approach to health and wellbeing was taken well beyond the specifics of the actual job being performed. When an organisation provides both supported employment and other services such as supported living and centre-based programs (leisure activity type programs), this coordination becomes somewhat easier. If there are multiple sites, then it is important that staff stay in communication with one another so that there is a common understanding of individual support requirements. Multiple services and strong relationships between support staff was considered an important element of promoting wellbeing:
*“I think we’re really lucky in terms of we have multiple sites and different supports, we’ve got our centre base, we’ve got our in-home support, supported living. We’re a very close organisation where all of the managers are constantly conversing over the phone, emailing. We all have very strong relationships. So we’re able to pick up the phone at any time and really mix in all of the services to make sure that we can put the wraparound supports to the person to get the best results … it’s not just about the employment, it’s looking at those other wraparound supports for them in the morning before they come to work and things like that”.*(Organisation 2, Staff 18)

Equally though, staff were able to coordinate across multiple services both within and outside their organisation if there were health and wellbeing issues that need attention:
*“There was an incident where he overdosed on his medication on the weekend. And it was a critical incident, where staff ended up temporarily taking his medication to another site. And temporarily we’re going to, under supervision, administer the medication. Whereas before, he was independently doing it, with prompts… So there’s a whole lot of coordination around that sort of stuff, so that he can continue working”.*(Organisation 2, Staff 18)

In the above example, the employment service needed to coordinate with the GP and family members to ensure that this situation was handled appropriately, and the person was able to continue working. Some organisations were able to provide a range of supports within the workplace with an example of counselling support:
*“It’s not the person’s intellectual disability, it’s not the person’s physical disability, quite often it’s that sort of psychosocial area that the person needs the most amount of support with. So having counsellors that can go into the workplace, that know the person really well, that can communicate with the person and support the person to self-advocate within an organisation, is huge for that person’s integration and mental health in the workplace”.*(Organisation 3, Staff 22)

Other organisations were able to find referrals as necessary based on their networks and understanding of the needs of the supported employee:
*“Obviously, that was when the alarm bells started going off and then eventually the participant responded and said that they weren’t really feeling okay mentally and we had to, obviously, have a little meeting and see what we could do. And we did get in contact with the parent and had to tell them about the urgency of it and how it was important to address it now. And then we referred them to some short-term immediate helplines services but also then, later down the track, we gave them some contacts for a counsellor and psychologist that they could use with their NDIS funding”.*(Organisation 1, Staff 14)

This holistic support in the workplace beyond just support with work tasks and routines was commented upon by supported employees. They acknowledged being supported in areas such as mental health, money and accommodation:
*“They’ve helped me a lot, with houses and everything in the past”.*(Organisation 1, Supported Employee 4)
*“They can help us out with outside problems as well”.*(Organisation 1, Supported Employee 1)

One of the consistent comments that underpins much of the holistic support was a culture and practice of constantly checking in with employees. There were examples of workplace routines for check-in, such as regular morning catch ups and also incidental conversations between support staff and supported employees to ensure that they were feeling ok that day:
*“I guess checking up on them, making sure that they’re okay and that they have a say and they’re contributing to what’s happening in the workplace as well to make sure that they feel valued as an employee. What we do is we have morning huddles, we call them. So it’s pretty much having a little bit of a team meeting at each of the social enterprises and talking about what’s happening for the day”.*(Organisation 2, Staff 17)

This was part of creating a safe work environment, which was highly valued by supported employees:
*“I think just everybody should be able to feel safe in the job”.*(Organisation 2, Supported Employee 14)

There are a range of things that contribute to wellbeing outside of the workplace. External factors can influence both wellbeing as well as work attendance and participation. Absenteeism or being late may be associated with a lack of external support. The health and wellbeing of supported employees might be at risk due to their current living arrangements. This can include issues with sleep and their safety and security. Having a strong support structure at work can pick these issues up and link to external supports as well as support activities within the workplace:
*‘… if someone comes in late every day, there could be other issues. We try and look at everything. It could be an issue where they’ve run out of NDIS funding, and they’re walking to work …Everything can lead into something else, and that’s what I tell my team. We always try and think, especially changing behaviours, what’s going on?’*(Organisation 1, Staff 3)

According to some staff interviewed, there was a sense that the supports provided were beyond what would normally be expected in a standard workplace. One Disability Enterprise staff member described examples of wrap around support which is an accepted culture within the organization.
*“They may not have support coordination. They may not have some core supports in their plan … There’s a gap. And our team do fill that. They assist with doctors’ appointments. They’ve been assisting people with vaccination appointments and getting prepared for that … So, there’s things that we do over and above … Our core purpose about having a positive impact on people’s lives is very much ingrained”.*(Organisation 1, Staff 1)
*“Well, first of all, we make sure that people have got lunch, and if they don’t, we ask why. ‘My ATM card’s not working.’ Well, you wouldn’t let somebody go without lunch, so you’d let them borrow $5.00 or $10.00, and hopefully you’ll get it back from their parents. What else do we do? We do welfare checks, like if there’s somebody that hasn’t—somebody reaches out to me regularly, but they haven’t for a while, so I will call to make sure they’re all right”.*(Organisation 1, Staff 16)

What becomes apparent through the data is that promoting wellbeing was a core focus of the workplace and that irrespective of the setting (accommodation, centre-based activities, workplace) there was a strong sense of providing holistic support and care.

### 3.7. Prioritising Social Connections and Respect

The last theme discussed relates to the relationships that supported employees have with each other and also with support staff. In respect to the relationships with support staff, it was considered important that people feel valued as employees and colleagues and that people are respected for what they can contribute. If supported employees feel that their contribution is acknowledged, then they feel part of the organisation and respected:
*‘they’re excellent here … You’re not looked at or judged like you have a disability here’*.(Organisation 1, Supported Employee 2)

In some Disability Enterprises, they have what is termed ‘Buddy’ or mentoring systems where a particular worker is allocated to a supported employee. The Buddy/Mentor helps with the orientation process and all the work-related procedures that need to be followed. It was suggested that a ‘side by side’ working approach rather than a supervision approach was the best method and that a ‘buddy system’ may be useful in open employment contexts as well. There would need to be funding and training available for a ‘buddy’ system to work effectively in open employment contexts according to the interview participants. These mentors/buddies are important in being able to provide a supportive environment that fosters positive mental health and wellbeing:
*‘I think that if crew feel safe and they’ve got that positivity around them, positive staff to encourage them where they need to be. I think just having those kinds of things in place, it is better for their mental health. And obviously, if their mental health is on track, then they’re going to be better at what they’re doing… “Okay, I feel emotionally good at work, I feel mentally good at work, I feel supported at work. I feel safe at work,” all those kinds of things reflect in the work that they do physically’.*(Organisation 1, Staff 8)

While the results have focused on the organisational structures and strategies that were seen as supporting positive health and wellbeing, there were also strategies to try and mitigate wellbeing challenges. One of these challenges was peer conflict and the importance of actively monitoring for relationships that are turning negative, including signs of conflict that could interfere with performance at work:
*“That’s definitely one we look out for and also just keeping an eye on how their relationships are going with other people in the workplace, because sometimes there can be conflict and that can obviously impact them”.*(Organisation 1, Staff 2)

Having a workplace with multiple sites and work roles was viewed as helpful in ensuring that people could be moved to avoid certain conflicts and being able to place people that connected well together.

## 4. Discussion

The results are consistent with previous research on the importance of strong relationships between managers, support staff and employees [29,40]. Strategies found amongst these case study organisations such as ‘checking-in’ about their wellbeing, mentoring and job coaching, and having a person-centred approach to planning, have also been found in previous research [29,41]. The results extend some of the previous research by uncovering some of the broader organisational characteristics that supported the health and wellbeing of people with an intellectual disability in the workplace. Of particular importance was being able to offer a variety of roles and different workplace environments such as quiet versus more social spaces. These themes reiterate previous research on how organisational design and space configuration of social enterprises can impact on the health and well-being outcomes of employees [42,43].

One of the criticisms of health promotion in workplace wellbeing is its focus on programmatic approaches rather than consideration of the design of the workplace itself [28]. This study highlights that how the workplace itself is designed with respect to use of space and the specific roles of support staff that are written into job descriptions, are integral to promoting the wellbeing of employees. Other features such as flexible work rosters and providing a diversity of roles and opportunities were also viewed as particularly important for mental health and overall wellbeing. The culture of the workplace itself was also pivotal with respect to adopting a holistic approach to wellbeing, which was evident in addressing a myriad of issues from medication, mental health episodes, balancing work with other social commitments, and coordination of different services.

These findings can assist in developing a set of standards and guide for promoting wellbeing in the workplace for people with an intellectual disability [12]. The recent Disability Royal Commission in Australia has recommended that disability enterprises need to reform and provide more inclusive and community focused roles [17]. There is currently a lack of definition and standards for workplace inclusion and wellbeing for workplaces employing people with an intellectual disability [12], and this research provides a starting point for the development of these standards. Given the diversity of industries, locations, and mix of employees of different workplaces, the challenge in developing standards and guides will be ascertaining standards of indicators that would be considered core elements versus those that could vary between organisations [44]. It has been previously noted that developing evaluation standards for social enterprises has a number of challenges [45], one of which is trying to determine which organisational features and strategies should be replicated [44].

Some of the findings from this study can also potentially be used in open employment settings that employ people with an intellectual disability. Transition rates from supported to open employment are very low in Australia and internationally [16,18], and the policy aspiration is that more people with intellectual disability be employed in open employment rather than supported employment [17]. The ambition is that school leavers with an intellectual disability are first considered for open employment roles [17]. Some of the strategies used in these organisations could potentially be replicated in open employment settings such as the buddy/mentor relationship and flexibility in roles and rosters. It is probably unrealistic to expect an open employer would be able to provide the same level of holistic support that the case study organisations can due to their deep industry experience and connections, which highlights the need for ongoing support for people as they transition to open employment roles [46].

The strength of this research was developing an understanding of the broader organizational characteristics that positively impacted on the health and well-being of people with an intellectual disability. While the study was limited to four case studies, these organisations were from diverse industries and locations. Also, a limitation was that the majority of interviews took place with one organisation, and hence the results might be biased to reflect the operation of this organisation. However, the results and themes were shared with key stakeholders from all organisations and there was consensus that these themes described well the overall approach to each organisation. More family members could also have been interviewed on their views on what constitutes a workplace that supports health and wellbeing. The strength of the study relates to its significant limitation in that we were specifically focusing on positive impacts and organisational features. The purpose of this research was to provide a starting point for developing a series of standards and guide for how to promote workplace health and wellbeing of people with an intellectual disability. The findings have provided a strong understanding of how the organisation can support mental health and wellbeing broadly, but there was less discussion on physical health and this needs to be an area of future research. The extent to which people with an intellectual disability experience the workplace as described in this study is also unknown and represents a significant gap that future research will need to address. The study did not address some of the negative aspects of work which exist for people with an intellectual disability [17]. Future quantitative research can also address the extent to which these organisational factors are related to health and wellbeing in the workplace.

## 5. Conclusions

The findings of this study revealed how organisational features and design characteristics such as the variable use of space to create quiet and social locations, having a variety of roles, customised training and development opportunities, and flexible work rosters contributed to the health and wellbeing of employees, in particular, mental and social wellbeing. The results also supported past research on the importance of quality relationships with support staff and managers. These findings can help in developing industry standards and guides for both supported and open employment settings, ensuring that people with an intellectual disability have a positive health and wellbeing experience in the workplace. Further research can explore how these organisational factors are related to health and wellbeing outcomes using quantitative methods and which of these organisational characteristics and strategies are important to replicate. Applying this research in practice, particularly in open employment contexts, could potentially play an important role in increasing inclusive employment experiences for people with an intellectual disability.

## Data Availability

The data presented in this study are available on request from the corresponding author due to privacy considerations.

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
