# Peer review of "Workplace Structures and Culture That Support the Wellbeing of People with an Intellectual Disability"

_ijerph, 2024, doi:10.3390/ijerph21111453_

Round 1

Reviewer 1 Report

Comments and Suggestions for Authors

15/10/2024

Review of the manuscript: Workplace structures and culture that support the health and wellbeing of people with an intellectual disability 

I appreciate the opportunity to review the manuscript - Workplace structures and culture that support the health and wellbeing of people with an intellectual disability

I find the manuscript a valuable qualitative research report which adds to the current state of knowledge on how to promote wellbeing of people with intellectual disabilities in workplace structures. 

In the process of reviewing I adopted the Critical Appraisal Skills Programme (Critical Appraisal Skills Programme UK: CASP checklists: https://casp-uk.net/images/checklist/documents/CASP-Qualitative-Studies-Checklist/CASP-Qualitative-Checklist-2018_fillable_form.pdf), which provides a ten-question checklist-based framework for assessing the quality and rigor of qualitative studies.

1)        Aims of the research 

The study aimed “to examine the organisational characteristics, structures, and cultural elements that contribute to positive health and wellbeing” (p. 1)

As such, I find it important and relevant. 

2)        Methods

The qualitative type of research addresses its aims. 

3)        Design

The research design was appropriate to address the aims of the research. 

4)        Recruitment

The process of recruitment could be explained in more detail. For example, how were potential participants encouraged to participate in the study? How many people refused to take part in the study? 

5)        Data collection

The chosen method is appropriate, as the authors have clearly explained the procedure and the data collection process, which involved individual interviews. However, providing details about the timing and settings of data collection would enhance transparency. Additionally, including examples of the interview questions would be beneficial.

More information about the participants is also recommended. The report could clarify how participants are referred to, as there is some inconsistency in the labeling. For instance, participants are sometimes referred to as "(Stage 1, Supported Employee 9)" and at other times as "(Organisation 2, Staff 17)" or "(Supported Employee 8)." It would be helpful to maintain consistency, such as always specifying the organization the participant represents. Additionally, the reason for referencing "stages" is not clear and could be better explained.

Finally, the report mentions that family members of supported employees participated in the study, but none of their perspectives are cited. Including at least one quotation from a family member would strengthen the report.

6)        Researcher role

There is not much information about the researchers’ own role and potential bias during data collection, including participants recruitment and choice of location. 

7)        Ethics

I do not have any doubts about ethical issues in this report. Participants gave an informed consent to participate in the study and the project was approved by a relevant ethics committee.

8)        Analysis

The steps of the analysis are well-described. It is clear how the categories were derived from the data. 

9)        Findings and Discussion

The findings are clearly presented and discussed. However, a minor issue is the absence of quotations from parents, as previously mentioned.

10)      Value

I find the study valuable, especially for practical reasons. As the authors state, the findings “can assist in developing a set of standards and guide for promoting wellbeing in the workplace for people with an intellectual disability” (p. 10).

Obviously, the manuscript does not require a proof-reading by a native speaker, but there are a few minor mistakes in writing, which can be easily corrected. Examples:

-              A sentence “These findings can help in developing industry standards and guides for both supported and open employment settings so that people with an intellectual disability a positive health and wellbeing experience in the workplace” could be made clearer, e.g. “These findings can help in developing industry standards and guidelines for both supported and open employment settings, ensuring that people with intellectual disabilities have a positive health and well-being experience in the workplace.”

-              In a sentence “As a credibility and integrity check, the findings the themes of the study were presented and discussed in meetings with some of the interview participants” either the words “the findings” or “the themes” should be removed to avoid redundancy.

-              In a sentence “Participants were provided with provided with a Plain Language Statement which was also available in Easy English and if required, the information was read to participants” the phrase “provided with” is repeated.

Author Response

Reviewer 1

I find the manuscript a valuable qualitative research report which adds to the current state of knowledge on how to promote wellbeing of people with intellectual disabilities in workplace structures.

Response: Thanks

In the process of reviewing I adopted the Critical Appraisal Skills Programme (Critical Appraisal Skills Programme UK: CASP checklists: https://casp-uk.net/images/checklist/documents/CASP-Qualitative-Studies-Checklist/CASP-Qualitative-Checklist-2018_fillable_form.pdf), which provides a ten-question checklist-based framework for assessing the quality and rigor of qualitative studies.

1)        Aims of the research 

The study aimed “to examine the organisational characteristics, structures, and cultural elements that contribute to positive health and wellbeing” (p. 1)

As such, I find it important and relevant. 

Response: Thanks

2)        Methods

The qualitative type of research addresses its aims. 

3)        Design

The research design was appropriate to address the aims of the research. 

4)        Recruitment

The process of recruitment could be explained in more detail. For example, how were potential participants encouraged to participate in the study? How many people refused to take part in the study? 

Response: Further detail about the process of selecting the actual case organisations, inclusion criteria of the interview participants, and also the process of recruitment of the interview participants has been provided. Detail was not collected on who refused to take part in the study as it was an open invitation across large organizations.

5)        Data collection

The chosen method is appropriate, as the authors have clearly explained the procedure and the data collection process, which involved individual interviews. However, providing details about the timing and settings of data collection would enhance transparency. Additionally, including examples of the interview questions would be beneficial.

Response: Further detail about the location of the interviews has been added. Example questions have been added.

More information about the participants is also recommended. The report could clarify how participants are referred to, as there is some inconsistency in the labeling. For instance, participants are sometimes referred to as "(Stage 1, Supported Employee 9)" and at other times as "(Organisation 2, Staff 17)" or "(Supported Employee 8)." It would be helpful to maintain consistency, such as always specifying the organization the participant represents. Additionally, the reason for referencing "stages" is not clear and could be better explained.

Response: We have added more detail about the participants. Thanks for pointing out the inconsistency in labelling quotes. The information about stages related to a previous draft about the timing of the data collection whereas all quotes should have referenced the organization which they now have.

Finally, the report mentions that family members of supported employees participated in the study, but none of their perspectives are cited. Including at least one quotation from a family member would strengthen the report.

Response: Further clarity of the numbers of participants has been provided in the methods section. There was only 1 family member. The interview with the family member was focused entirely on feedback on the process of transitioning to open employment and some of the policy barriers that prevented a successful transition. This was part of the other aim of the overall study on employment transitions but the data from this interview was not relevant for this paper. The lack of family members has been noted as a further limitation of the paper.

6)        Researcher role

There is not much information about the researchers’ own role and potential bias during data collection, including participants recruitment and choice of location.

Response: Further information has been added as requested.

7)        Ethics

I do not have any doubts about ethical issues in this report. Participants gave an informed consent to participate in the study and the project was approved by a relevant ethics committee.

8)        Analysis

The steps of the analysis are well-described. It is clear how the categories were derived from the data. 

9)        Findings and Discussion

The findings are clearly presented and discussed. However, a minor issue is the absence of quotations from parents, as previously mentioned.

Response: The absence of material from parents was added a further limitation of the study.

10)      Value

I find the study valuable, especially for practical reasons. As the authors state, the findings “can assist in developing a set of standards and guide for promoting wellbeing in the workplace for people with an intellectual disability” (p. 10).

 Response: Thanks for this feedback

Obviously, the manuscript does not require a proof-reading by a native speaker, but there are a few minor mistakes in writing, which can be easily corrected. Examples:

-              A sentence “These findings can help in developing industry standards and guides for both supported and open employment settings so that people with an intellectual disability a positive health and wellbeing experience in the workplace” could be made clearer, e.g. “These findings can help in developing industry standards and guidelines for both supported and open employment settings, ensuring that people with intellectual disabilities have a positive health and well-being experience in the workplace.”

-              In a sentence “As a credibility and integrity check, the findings the themes of the study were presented and discussed in meetings with some of the interview participants” either the words “the findings” or “the themes” should be removed to avoid redundancy.

 -              In a sentence “Participants were provided with provided with a Plain Language Statement which was also available in Easy English and if required, the information was read to participants” the phrase “provided with” is repeated.

 Response: Thanks for pointing out these sentences. They have been fixed.

Reviewer 2 Report

Comments and Suggestions for Authors

This manuscript presents a study on wellbeing in workplaces for people with intellectual disabilities. The study is well written and motivated. The method should be strengthened by adding more information about the analyses. The results could be organised differently such that is clear which themes and descriptions come from organisations/staff, and which come from employees and family. 

Introduction

Lines 57-61: in an introduction, the results should not already be mentioned. Instead, at this place, it seems logical to clarify the research question a bit more. If interviews are being held with both employees (with ID) and managers/stakeholders, what are you investigating with each of these groups? 

The text about inclusive employment for people with ID (lines 63-80) is not entirely clear to me. What do you mean exactly by ‘inclusive’ and what is meant by ‘open employment’? Does open employment mean an employment at companies where people with and without ID are employed? From the text I get the feeling that open employment is somehow preferred. Is this correct and can you motivate why this is the case?

At the end of the introduction, a few lines on the research question(s) and the approach would be helpful. Up to this point, all the reader gets is the aim (final sentence) and the fact that you will perform interviews. 

Method

It becomes clear that you focused on health and wellbeing. Therefore, these terms need to be defined (in the introduction). 

The interview questions for staff and family focusses on health and wellbeing, but from what I read, the interview question for employees with ID focus on wellbeing but not on health. How do you justify this?

It would be good to add a table with participant information. 

Is it possible to add the interview guides in supplementary material or appendix? Some more information about the interviews is required. Who performed the interviews? Were they structured or unstructured? Did the interviewer ask follow-up questions? How long were the interviews? Were the interviews transcribed? Etc.

To address the validity and reliability of the research, more details need to be given on the process of analysis. Did you perform open coding as a first stage? How many researchers were involved in coding? Did you develop a code book? Did you calculate inter-coder-reliability? Etc. 

The interviews with the group of employees (with ID) focused on different questions than the interviews with staff and family.  Did you therefore analyse this group separately from the other group? If yes, please provide details of the separate analysis (and combination of the results after these two stages). If no, why not?

You write that you did a member check with “some of the interview participants”. Why not with all of them?

Results

In the description of the themes, the text seems to convey the ideas/opinions of the organisations. This is not made explicit, which makes it confusing to read. Elements in the text like “was considered an important element”, seem to mean ‘considered important by the staff members of the organisation’. This connects to my remark in the method about doing a separate analysis of the employees on the one hand and the staff/organisations on the other hand (and perhaps family members as a third group). The way the text reads now, it is not clear what elements in the descriptions are coming from the staff and which from the employees. However, the reader gets the feeling that most of it comes from the organisations. I would recommend presenting the results separately for the different groups, or at least be more explicit about what information is coming from whom. 

Some more specific feedback:

What does ‘Stage’ mean? Does this have to do with the stage of analysis or something else? Please explain. 

In the first theme (Diversity of Roles and Opportunities) the diversity of role and supporting change is stressed. However, the supporting quote from an employee says that he/she likes his/her main role as front of the house. Why is this a supporting quote for this theme?

The text describing the themes (apart from the quotes) are written as they are being stated by the organisations (instead of by the employees). For example: “While providing customised tasks was seen as pivotal it was also mentioned that a strong sense of purpose of the role itself was important.” Sometimes, the text even feels as a (promotion) line from the organisation itself: “Having a large organisation with different types of opportunities was very important in being able to task match.” What does this mean? Was this true for all four organisations involved?

How does theme 5 (Offering a Range of Workplace Environments) differ from theme 1 (Diversity of Roles and Opportunities)? They seem very connected to me, especially if I look at the first quote of each theme.  

“Overall, the culture of the organisations is going beyond that which would be expected within a workplace in the support of their social goals and the people they work with.” This is another example of a sentence that sound like promoting the organisations. Please be more objective. Did some staff-member say this? Then use quotation marks. It also raises additional questions, like: what are the social goals and what is expected in a workplace?

In themes 6 and 7, I don’t see any quotes from employees. Did these themes not emerge from their interviews?

Does the final quote of theme 3 (about contributing and being appreciated) does not fit better with theme 7?

From the description of the focus of the interviews I learned that there would be a focus on wellbeing and health. However, from the results it is not apparent health was discussed, apart from ‘mental health’ and one incident with medication. In the introduction, it is suggested that the topic is also about physical health (healthy food, physical exercise, etc.), or did I misunderstand? This would be good to clarify. 

One of the questions to the employees with ID was what they liked and did not like about their job. The results, however, do not describe anything about things that they did not like.

Discussion

The fact that the data is largely coming from only one organization is a major limitation of the study. In the discussion you mention that the other stakeholders agreed about the themes (after the analysis?). Did you also get support for these themes during the analysis?

The language is fine, but please do a careful proofread to eliminate typos and non-grammatical sentences (as a result of editing). 

Author Response

Reviewer 2

This manuscript presents a study on wellbeing in workplaces for people with intellectual disabilities. The study is well written and motivated. The method should be strengthened by adding more information about the analyses. The results could be organised differently such that is clear which themes and descriptions come from organisations/staff, and which come from employees and family. 

Introduction

Lines 57-61: in an introduction, the results should not already be mentioned. Instead, at this place, it seems logical to clarify the research question a bit more. If interviews are being held with both employees (with ID) and managers/stakeholders, what are you investigating with each of these groups?

Response: Providing a snapshot of the results both in the abstract and at this point in the introduction is a style issue that we have used in a number of papers as a signpost to the readers. However, given this feedback we have removed mention of the results here. Instead we have provided further detail about the research questions, choice of participants, and the definitions used of health and wellbeing.

The text about inclusive employment for people with ID (lines 63-80) is not entirely clear to me. What do you mean exactly by ‘inclusive’ and what is meant by ‘open employment’? Does open employment mean an employment at companies where people with and without ID are employed? From the text I get the feeling that open employment is somehow preferred. Is this correct and can you motivate why this is the case?

Response: Further clarity and detail has been provided.

At the end of the introduction, a few lines on the research question(s) and the approach would be helpful. Up to this point, all the reader gets is the aim (final sentence) and the fact that you will perform interviews.

Response: Further detail has been added.

Method

It becomes clear that you focused on health and wellbeing. Therefore, these terms need to be defined (in the introduction).

Response: This has been added.

The interview questions for staff and family focusses on health and wellbeing, but from what I read, the interview question for employees with ID focus on wellbeing but not on health. How do you justify this?

Response: Further detail has been added about the questions. This was a semi-structured interview schedule so questions were adapted during the course of each interview. The questions for people with intellectual disability needed to be modified to suit their ability levels so they were kept conceptually simple to what they liked about the job and what supports they needed.

It would be good to add a table with participant information.

Response: Further detail has been added about the participants. However, we did not collect any personal information consistent with the ethics approval and thus there would be nothing further to add in a table.

Is it possible to add the interview guides in supplementary material or appendix? Some more information about the interviews is required. Who performed the interviews? Were they structured or unstructured? Did the interviewer ask follow-up questions? How long were the interviews? Were the interviews transcribed? Etc.

Response: Further detail has been added in the method section.

To address the validity and reliability of the research, more details need to be given on the process of analysis. Did you perform open coding as a first stage? How many researchers were involved in coding? Did you develop a code book? Did you calculate inter-coder-reliability? Etc.

Response: Further detail has been added.

The interviews with the group of employees (with ID) focused on different questions than the interviews with staff and family.  Did you therefore analyse this group separately from the other group? If yes, please provide details of the separate analysis (and combination of the results after these two stages). If no, why not?

Response: Further detail has been added in the introduction, method and results about how the interviews of the different groups were connected together.

You write that you did a member check with “some of the interview participants”. Why not with all of them?

Response: Further clarification has been provided. It was not possible to conduct a conceptual check of the data with the supported employees due to their ability levels.

Results

In the description of the themes, the text seems to convey the ideas/opinions of the organisations. This is not made explicit, which makes it confusing to read. Elements in the text like “was considered an important element”, seem to mean ‘considered important by the staff members of the organisation’. This connects to my remark in the method about doing a separate analysis of the employees on the one hand and the staff/organisations on the other hand (and perhaps family members as a third group). The way the text reads now, it is not clear what elements in the descriptions are coming from the staff and which from the employees. However, the reader gets the feeling that most of it comes from the organisations. I would recommend presenting the results separately for the different groups, or at least be more explicit about what information is coming from whom.

Response: Thanks for this very helpful feedback we agree this was not clearly presented in the results. Further detail has been provided in the introduction, method and start of the results about the specific purpose of the interviews. The main source was the staff and manager interviews as we wanted to understand concepts around how organizational structures and processes influenced health and wellbeing. These kinds of concepts are somewhat challenging for people with an intellectual disability to discuss. The interviews with people with an intellectual disability formed more of a check to see if the themes discussed by staff and managers was consistent in what the supported employees liked about their job and what supports they felt were valuable using interview questions that were within their ability level. We want to present the results as the key themes related to the organizational structures and processes that influenced health and wellbeing which is the purpose of the study rather than separate out the different interview groups.

Some more specific feedback:

What does ‘Stage’ mean? Does this have to do with the stage of analysis or something else? Please explain.

Response: Thanks for pointing this out, that is a mistake to still include a reference to stage. It was from an earlier draft of the paper about the timing of the data collection and should not have been in the final version. This has now been removed and greater consistency applied to how we label the quotes.

In the first theme (Diversity of Roles and Opportunities) the diversity of role and supporting change is stressed. However, the supporting quote from an employee says that he/she likes his/her main role as front of the house. Why is this a supporting quote for this theme?

Response: Further data has been provided for this quote to give more context. And an additional quote was also added to help clarify this point.

The text describing the themes (apart from the quotes) are written as they are being stated by the organisations (instead of by the employees). For example: “While providing customised tasks was seen as pivotal it was also mentioned that a strong sense of purpose of the role itself was important.” Sometimes, the text even feels as a (promotion) line from the organisation itself: “Having a large organisation with different types of opportunities was very important in being able to task match.” What does this mean? Was this true for all four organisations involved?

Response: Thanks for pointing this out. We have been clearer now on attributing quotes and the text explaining the themes to the interview participants.

How does theme 5 (Offering a Range of Workplace Environments) differ from theme 1 (Diversity of Roles and Opportunities)? They seem very connected to me, especially if I look at the first quote of each theme. 

Response: They are similar but distinct. The first theme was about diversity of work tasks and theme 5 about variety in physical workspaces such as quiet rooms being made available. Further clarification about this point has been made.

“Overall, the culture of the organisations is going beyond that which would be expected within a workplace in the support of their social goals and the people they work with.” This is another example of a sentence that sound like promoting the organisations. Please be more objective. Did some staff-member say this? Then use quotation marks. It also raises additional questions, like: what are the social goals and what is expected in a workplace?

Response: This related directly to the next quote where the interview respondent used the phrase ‘over and above.’ However, we agree with this critique that the sentence was not well phrased and needed to be more objective and it has been changed.

In themes 6 and 7, I don’t see any quotes from employees. Did these themes not emerge from their interviews?

Response: There was a quote from an employee in section 7 but not any in section 6. Thanks for pointing this out this. The interviewees with intellectual disability were not as articulate in explaining some of these concepts but not having quotes from supported employees in theme 6 was an oversight and we have rectified this. Thanks again for your help in noticing this.

Does the final quote of theme 3 (about contributing and being appreciated) does not fit better with theme 7?

Response: We agree this quote could also have been used in theme 7. The reason we have chosen to use it in theme 3 relates to the challenge of task modification in ensuring that there is still purpose to the role. There is a history within the industry of using day programs and sheltered workshops to keep people’s days occupied without any real sense of purpose. So when task matching it is important that a sense of purpose is there. We have added another quote to theme 7 about sense of purpose though to address this comment.

From the description of the focus of the interviews I learned that there would be a focus on wellbeing and health. However, from the results it is not apparent health was discussed, apart from ‘mental health’ and one incident with medication. In the introduction, it is suggested that the topic is also about physical health (healthy food, physical exercise, etc.), or did I misunderstand? This would be good to clarify.

Response: Yes this was very much the intention of the interviews to discuss the broad concept of health and as mentioned previously we have now defined health and wellbeing in the introduction and added more detail about the interview questions in the method. That physical health did not come up much is an interesting finding in itself. There was some mention of providing lunch but nothing about making sure it was healthy food. We have added this as a limitation of the study which further research would need to explore.

One of the questions to the employees with ID was what they liked and did not like about their job. The results, however, do not describe anything about things that they did not like.

Response: This was mentioned as a limitation of the study. The specific wording of the interview questions was added to the method section.

Discussion

The fact that the data is largely coming from only one organization is a major limitation of the study. In the discussion you mention that the other stakeholders agreed about the themes (after the analysis?). Did you also get support for these themes during the analysis?

Response: Yes this was acknowledged as a major limitation but as described the set of themes were verified by stakeholders from all four organisations and this process has been described in the paper.

The language is fine, but please do a careful proofread to eliminate typos and non-grammatical sentences (as a result of editing). 

Response: We have conducted a further edit and addressed these concerns.

Round 2

Reviewer 2 Report

Comments and Suggestions for Authors

The manuscript has improved, and the authors have made significant changes (especially given the short period for revision). Many points of feedback have now been clarified. Still, there are a few things I would like to point out.  

The title, abstract and introduction all suggest that this study is about health, which is not the case. The first sentence of the abstract starts with “There is little research on health promoting workplace settings”. The introduction mentions that “this population cohort have higher rates of a number of health conditions including diabetes [3] and obesity [3,4]. People with intellectual disability have recorded lower levels of physical activity and less healthy diets”. Furthermore, the words “promoting health” are used in several sentences. This all suggests that the presents study is about investigating the promotion of physical health in workplace environments. However, this is not the case. From the interview questions, we learn that the questions are not about providing health-related programs or otherwise promoting physical health, but about supporting (mental) health and well-being. 

You may argue that you asked about health which could also be interpreted as being physical health, but as the results section is not about physical health promotion, I would suggest that you rewrite the title, abstract and introduction such that these are aligned with the topic/findings of your study. 

Some more detailed comments:

Lines 70-92: The terms inclusive employment and open employment are still not entirely clear to me. Is it possible to provide a definition with a reference? This would help (not only me, I assume). It seems that these terms are used interchangeably and are meaning (almost) the same. Please clarify. 

lines 193-195: “Given the limited research on health and wellbeing in the workplace for people with an intellectual disability, there was no particular analytic frame that could be used and hence a thematic analysis was undertaken”. 

This sounds as if a thematic analysis was the least preferred option and you chose this only because nothing else could be done, but this is probably not what you mean. Could you clarify what you do mean, instead? Do you mean that because of the limited research not deductive coding could be used and hence an inductive/open coding approach was used?

Only in the Results section, the reader learns that the interviews with the partner organisations and family member are not used for the analysis in this study. Please mention this information earlier, in the Method, and report there (in the Method) on the correct number of participants (= the participants that contributed to the data for this specific study). It is fine (and helpful) to read that this research is part of a larger study, but after you have stated this (in the Method), all further details provided should focus on this particular study that you are reporting on. 

You wrote: “It was not possible to conduct a conceptual check of the data with the supported employees due to their ability levels.”

If you did not manage to do this, please mention this in the discussion as a limitation of the research. Just for your information: there are examples from other studies where the researchers managed to do this. 

You wrote: “The interviews with people with an intellectual disability formed more of a check to see if the themes discussed by staff and managers was consistent in what the supported employees liked about their job and what supports they felt were valuable”. 

If this is the case, it would be good to make more explicit in the Methods section. So do I understand correctly, that you first analysed the interviews with the staff, and then looked into the interviews with the employees whether their quotes supported the themes you have found from the staff interviews?

You mentioned that “We have been clearer now on attributing quotes and the text explaining the themes to the interview participants.”

With “the interview participants” you mean both staff and employees, right? So with adding this (e.g. in line 326-327) you mean that this information came from both groups? If so, the reader would like to see these statements supported by quotes from both groups. 

Response: Yes this was very much the intention of the interviews to discuss the broad concept of health and as mentioned previously we have now defined health and wellbeing in the introduction and added more detail about the interview questions in the method. That physical health did not come up much is an interesting finding in itself.

The interview questions show that you asked about how they support health and well-being, but not what kind of health promotion programs they offer (if any). Also, ‘health’ is mentioned only in the first question together with the word well-being, which makes it easy to interpret ‘health’ as being mental health. Therefore, seeing the interview questions, I don’t find it surprising that the focus of the answers was on well-being instead of physical health. If you really wanted to investigate about physical health as well, why didn’t you ask for this specifically?

There are several typos/ not well flowing sentences (e.g. lines 62, 68, 169,196, etc). Please proofread the improved/final version carefully. 

Author Response

Reviewer 2

The manuscript has improved, and the authors have made significant changes (especially given the short period for revision). Many points of feedback have now been clarified. Still, there are a few things I would like to point out.  

The title, abstract and introduction all suggest that this study is about health, which is not the case. The first sentence of the abstract starts with “There is little research on health promoting workplace settings”. The introduction mentions that “this population cohort have higher rates of a number of health conditions including diabetes [3] and obesity [3,4]. People with intellectual disability have recorded lower levels of physical activity and less healthy diets”. Furthermore, the words “promoting health” are used in several sentences. This all suggests that the presents study is about investigating the promotion of physical health in workplace environments. However, this is not the case. From the interview questions, we learn that the questions are not about providing health-related programs or otherwise promoting physical health, but about supporting (mental) health and well-being. 

You may argue that you asked about health which could also be interpreted as being physical health, but as the results section is not about physical health promotion, I would suggest that you rewrite the title, abstract and introduction such that these are aligned with the topic/findings of your study.

Response: Thanks for this feedback. Yes the intent of the research and the interview schedule was about all facets of health but you are correct that mental and social wellbeing were the main topics discussed and analysed. And as such the title, abstract and introduction needs to reflect this. We have made these changes as suggested to reflect the main emphasis of the paper.

Some more detailed comments:

Lines 70-92: The terms inclusive employment and open employment are still not entirely clear to me. Is it possible to provide a definition with a reference? This would help (not only me, I assume). It seems that these terms are used interchangeably and are meaning (almost) the same. Please clarify. 

Response: This has been provided.

lines 193-195: “Given the limited research on health and wellbeing in the workplace for people with an intellectual disability, there was no particular analytic frame that could be used and hence a thematic analysis was undertaken”. 

This sounds as if a thematic analysis was the least preferred option and you chose this only because nothing else could be done, but this is probably not what you mean. Could you clarify what you do mean, instead? Do you mean that because of the limited research not deductive coding could be used and hence an inductive/open coding approach was used?

Response: Thanks yes this was poorly expressed and has been edited as suggested.

Only in the Results section, the reader learns that the interviews with the partner organisations and family member are not used for the analysis in this study. Please mention this information earlier, in the Method, and report there (in the Method) on the correct number of participants (= the participants that contributed to the data for this specific study). It is fine (and helpful) to read that this research is part of a larger study, but after you have stated this (in the Method), all further details provided should focus on this particular study that you are reporting on. 

 Response: This has been further clarified as suggested.

You wrote: “It was not possible to conduct a conceptual check of the data with the supported employees due to their ability levels.”

If you did not manage to do this, please mention this in the discussion as a limitation of the research. Just for your information: there are examples from other studies where the researchers managed to do this.

Response: In your first review you asked why we did not conduct a credibility and integrity check with all of the participants and our response was as mentioned above. To provide further clarity of the overall study, there were workshops with both staff and supported employees on a model we developed explaining the transition from supported to open employment. Thus, we did in this project and have in all our work included people with intellectual disabilities in various stages of the research process, including feeding back and refining research themes. For this particular paper the topic was on the organizational structures and processes that managers and staff were devising and implementing to support health and wellbeing. Thus, the themes themselves more related to the work of senior managers and staff within the organization and that is who we conducted this validity and integrity check. To draw attention to this as a limitation of the paper would not reflect our overall approach to this study across the various topics that we researched.

You wrote: “The interviews with people with an intellectual disability formed more of a check to see if the themes discussed by staff and managers was consistent in what the supported employees liked about their job and what supports they felt were valuable”. 

If this is the case, it would be good to make more explicit in the Methods section. So do I understand correctly, that you first analysed the interviews with the staff, and then looked into the interviews with the employees whether their quotes supported the themes you have found from the staff interviews?

Response: Our response could have been better phrased. The term ‘more of a check’ was not meant to be taken literally. We did analyse the data at the same time and all interviews were equally important in developing the themes. It is just given the analytic requirement of interview participants to be able to conceptually link organizational structures and processes to health and wellbeing outcomes, staff and managers were more easily able to articulate those connections. As an example, supported employees were able to tell us what was important such as variety but it was the staff and managers which were able to articulate how the structure and operation of the organization itself was able to provide such variety.

You mentioned that “We have been clearer now on attributing quotes and the text explaining the themes to the interview participants.”

With “the interview participants” you mean both staff and employees, right? So with adding this (e.g. in line 326-327) you mean that this information came from both groups? If so, the reader would like to see these statements supported by quotes from both groups.

Response: Yes that is correct. We have moved a quote around to reflect this.

You wrote: Yes this was very much the intention of the interviews to discuss the broad concept of health and as mentioned previously we have now defined health and wellbeing in the introduction and added more detail about the interview questions in the method. That physical health did not come up much is an interesting finding in itself.

The interview questions show that you asked about how they support health and well-being, but not what kind of health promotion programs they offer (if any). Also, ‘health’ is mentioned only in the first question together with the word well-being, which makes it easy to interpret ‘health’ as being mental health. Therefore, seeing the interview questions, I don’t find it surprising that the focus of the answers was on well-being instead of physical health. If you really wanted to investigate about physical health as well, why didn’t you ask for this specifically?

Response: We did have specific prompts about physical health and there were interview answers about medication, food and nutrition, and physical activity. The whole focus of the topic though was how the organization itself (its operation, procedures, processes etc.) supported health and wellbeing. We were not interested in whether they were providing programs such as education programs on physical activity or nutrition. We were interested in the actual organization design itself and the respondents did not really provide any information about how the organizational design, processes etc support physical activity and nutrition. This was an acknowledged limitation of the study.

There are several typos/ not well flowing sentences (e.g. lines 62, 68, 169,196, etc). Please proofread the improved/final version carefully.

Response: Thanks for picking these up we have corrected these errors.